# Dyeing Property Improvement of Madder with Polycarboxylic Acid for Cotton

**DOI:** 10.3390/polym13193289

**Published:** 2021-09-26

**Authors:** Xiaoyu Cai, Hong Li, Li Zhang, Jun Yan

**Affiliations:** Department of Textile and Material Engineering, Dalian Polytechnic University, Dalian 116034, China; caixiaoyu9924@163.com (X.C.); zhangli930705@163.com (L.Z.); yanjun@dlpu.edu.cn (J.Y.)

**Keywords:** natural madder dye, cotton fabric, polycarboxylic acid, cross-linking dyeing, esterification reaction

## Abstract

Cotton fabrics were dyed with the madder and compounds of citric acid (CA) and dicarboxylic acids [tartaric acid (TTA), malic acid (MLA), succinic acid (SUA)] as cross-linking agents and sodium hypophosphite (SHP) as the catalyst. The molecular structures and crystal structures of the dyed cotton fabrics were analyzed using Fourier-transform infrared spectroscopy (FTIR) and X-ray diffractometry (XRD), respectively. The results showed that the polycarboxylic acids esterified with the hydroxyl groups in the dye and cellulose, respectively, and the reaction mainly occurred in the amorphous region of the cotton fabric. Compared with the direct dyed cotton fabric, the surface color depth (*K/S*) values of the CA, CA+TTA, CA+MLA, CA+SUA cross-linked dyed cotton fabrics increased by approximately 160%, 190%, 240%, 270%, respectively. The CA+SUA cross-linked dyed cotton fabric achieved the biggest *K/S* value due to the elimination of the negative effect by α-hydroxyl in TTA and MLA on esterification reaction, and the cross-linked dyed cotton fabrics had great levelness property. The washing and rubbing fastness of the cross-linked cotton fabrics were above four levels. The light resistance stability and the antibacterial property of the cross-linked dyed cotton fabrics was obviously improved. The sum of warp and weft wrinkle recovery angle (WRA) of the CA+SUA cross-linked dyed cotton fabric was 55° higher than that of raw cotton fabric, and its average UV transmittance for UVA was less than 5% and its UPF value was 50+, showing a great anti-wrinkle and anti-ultraviolet properties.

## 1. Introduction

Most natural dyes are extracted from plant roots, stems, leaves, and flowers, and they are widely used in fabric dyeing due to the advantages of large reserves and rich colors [1]. Compared with the synthetic dyes, natural dyes have many excellent characteristics including low toxicity, anti-oxidation, anti-bacterial and anti-ultraviolet properties [2,3]. With the enhancement of environmental protection awareness, the development and utilization of natural dyes have been paid more and more attention by some researchers.

Natural madder dye is derived from rhizomes of the madder plant and belongs to anthraquinone structure dye [4]. As the main pigment of madder dye, Alizarin (1,2-dihydroxy anthraquinone) has a low affinity for the cotton fabric. Therefore, its direct dyeing property on the cotton fabric is poor. Alizarin has active α-hydroxyl, which can form complex structures with metal ions [5]. Normally, some metal ions are used as mordants to improve the surface color depth (*K/S*) value and color fastness of the dyed cotton fabric by madder, since the complex structures by the dye-mordant-cotton fabric make the dye be deposited on the cotton fabric in the mordant dyeing process. However, the heavy metals remain in the cotton fabric and dyeing effluent affect human skin health and cause some environmental problems [6]. In order to obtain ecological textiles, it is necessary to pursue a new eco-friendly dyeing method.

Cotton fabrics are widely used as daily textiles due to their hygroscopic, breathable and skin-friendly properties. A large number of active hydroxyl groups in the cellulose chain are easy to react with various cross-linking agents and obtain excellent functional textiles [7]. Polycarboxylic acids are usually used as formaldehyde-free cross-linking agents to enhance the wrinkle resistance property of the cotton fabric [8]. Research shows the cross-linking process of the polycarboxylic acid and cellulose is that the two adjacent carboxyl groups generate a five-membered cyclic anhydride intermediate under the conditions of catalyst and heating, and then reacts with the hydroxyl group on the cellulose chain to form the ester bond [9,10]. Among the many polycarboxylic acids, 1,2,3,4-butanetetracarboxylic acid (BTCA) is regarded as the most effective cross-linking agent [11]. However, the excessively high cost of the BTCA has prevented its application [12]. The citric acid (CA) is a cheap and environment-friendly finishing agent for the cotton fabric [13]. But, compared with the BTCA, fewer carboxyl groups in its molecular structure and the negative effect of α-hydroxyl in CA on the esterification reaction, so the cross-linking efficiency for the cotton fabric is reduced [14]. In order to improve the cross-linking efficiency of polycarboxylic acid for the cotton fabric, CA mixes with the BTCA to finish the cotton fabric [15], and results show that the wrinkle resistance of the cotton fabric after cross-linking is better than that of CA alone. However, the cost of this cross-linking process is also higher [16].

It is proved that the compound of polycarboxylic acids can create a synergistic effect during cross-linking cotton fabric [15,16]. But there are few reports on the compound of polycarboxylic acids applied on dyeing cotton fabric with natural dye. In the paper, the compound of dicarboxylic acid [tartaric acid (TTA), malic acid (MLA), succinic acid (SUA), the structures are shown in Figure 1] and CA as cross-linking agents were added to the dyeing solution of the cotton fabric by madder. The multiple esterification sites generated during the cross-linking dyeing process, and enhanced the cross-linking degree between the cotton fabric, polycarboxylic acid and dye, and improved the dyeing property of the cotton fabric. The morphology and structure of the cotton fabrics before and after dyeing were characterized, the cross-linking dyeing reaction mechanism was analyzed, the differences of color characteristic values, *K/S* values and color fastness to washing, rubbing and light were discussed, and the anti-wrinkle, anti-ultraviolet properties and the antibacterial property of the dyed cotton fabric were studied. It was looking forward to obtain the multifunctional cotton-based textile while improving the dyeing properties of cotton fabric with madder.

## 2. Experimental

### 2.1. Materials

Bleached cotton fabric (with 27 ends per cm and 22 picks per cm, Shan Dong Ruyi Technology Company, Shan Dong, China). Natural madder dye (solid commercial dyes, mainly component alizarin CAS: 70-48-0, alizarin percentage composition was 33%, Qing Dao Donghu Biological Material Co., Ltd., Shan Dong, China). Sodium carbonate (Na_2_CO_3_) and aluminum potassium sulfate (KAl(SO_4_)_2_·6H_2_O) (Liao Ning Xinxing Reagent Co., Ltd., Liao Ning, China). CA, TTA, MLA, SUA, sodium hypophosphite (SHP) and sodium hydroxide (NaOH) (Shang Hai Macklin Biochemical Co., Ltd., Shang Hai, China). Beef extract, peptone, nutrient agar, Escherichia coli (*E. coli*) (ATCC25922) and Staphylococcus aureus (*S. aureus*) (ATCC6538) (biological agents, Shanghai Guduo Biological Technology Co., Ltd., Shanghai, China). All reagents were of analytical grade.

### 2.2. Dyeing Methods

#### 2.2.1. Direct Dyeing

The cotton fabric was dyed with the madder dye concentration of 5% on weight of fabric (o.w.f.) at a ratio of 50:1 (the same below). The dyeing process was started at 30 °C. Then, it reached 90 °C during 20 min and kept for 30 min. After the dyed cotton fabric was padded with two dips and two nips [wet pick up (80% ± 1%), the same below], and it was washed with the running tap water and dried at room temperature (20–25 °C).

#### 2.2.2. Mordant Dyeing

The dye solution was prepared by adding 5% (o.w.f.) of the madder dye and alum mordant, respectively. And then the mordant dyeing was finished according to the direct dyeing process.

#### 2.2.3. Cross-Linked Dyeing

The cross-linked dyeing process was carried out using four types of cross-linking agents containing single CA, and compounds of CA with TTA, CA with MLA, and CA with SUA. The dye solution for a single CA cross-linking contained 5% (o.w.f.) of the madder dye, CA (1%) and SHP (6%), whereas the dye solution for CA mixed with dicarboxylic acid as a cross-linking agent contained 5% (o.w.f.) of the madder dye, CA (1%), and dicarboxylic acid with a molar ratio of 1:1 to CA, and SHP (6%). The dyeing process was started at 30 °C and reached 90 °C during 20 min and kept for 30 min, and then the dyed cotton fabric was padded with two dips and two nips, precured at 70 °C for 15 min and cured at 160 °C for 3 min. The cured cotton fabric was washed in deionized water to remove the unreacted acid, and then it was treated with a 0.1 mol/L NaOH solution at room temperature for 3 min to convert the unreacted carboxyl to carboxylate. Finally, the dyed cotton fabric was washed with running tap water and dried at room temperature.

### 2.3. Color Measurement

*L**, *a**, *b**, and *C** values of the dyed cotton fabrics were measured (the Color-Eye 7000A measurement instrument, Konica Minolta (China) Investment Co., Ltd., Shanghai, China) with a D65 light source and the 10° viewing angle. At the maximum absorption wavelength (420 nm), the dyed cotton fabrics *K/S* values were calculated [Equation (1)] [17].
(1)$$K/S=\frac{{\left(1-R\right)}^{2}}{2R}$$

On the basis of measured CIE color measurement, the color differences (Δ*E*) of the dyed cotton fabrics were calculated [Equation (2)]. The Δ*E* value is used to analyze the levelness of the dyed cotton fabric. The smaller the Δ*E* value, the better leveling property of the cotton fabric [18].
(2)$$\Delta E=\sqrt{\Delta L^{2}+\Delta a^{2}+\Delta b^{2}}$$

Based on CIE measurement, if *a** and *b** are both more or less than 0, the hue angle (*h°*) values of the dyed cotton fabrics were calculated [Equation (3)]. If one of the *a** and *b** values was greater than 0 and the other was less than 0, the *h°* values were calculated [Equation (4)]. Color characteristics and *K/S* values were measured thrice and took arithmetic average value [19].
(3)h=Arctan(b*/a*)
(4)h=360+Arctan(b*/a*)

### 2.4. Color Fastness

The washing fastness of the dyed cotton fabrics was measured according to AATCC-61-2007 (SW-12A, Wenzhou Da Rong Textile Equipment Co. Ltd., Wenzhou, China), and rubbing fastness was measured according to AATCC-8-2007 (Y571B, Wenzhou Da Rong Textile Equipment Co. Ltd., Wenzhou, China), and light fastness was measured according to AATCC-163-2014 (YG611S, Wenzhou Da Rong Textile Equipment Co. Ltd., Wenzhou, China).

### 2.5. Structural Characterization and Wearing Property

The KBr tablet method was used to analyze molecular structure of the cotton fabric using Fourier-transform infrared spectroscopy (FTIR) (FT-IR-650, Platinum Elmer Instruments Co., Ltd., Tianjin, China). The resolution was 4 cm^−1^, scan times were 100 times, and wavenumber ranged from 4000 to 500 cm^−1^. Crystal structures of the cotton fabrics were analyzed using X-ray diffractometer (XRD) (XRD-6100, Shimadzu Corporation, Xianggang, China), the diffraction source was Cu, and the scanning speed was 2 °/min, the range from 10°to 40°.

The wrinkle recovery angles (WRA) of the cotton fabrics were measured according to AATCC 66-2008 (Fabric Wrinkle Recovery, Recover Angle Method, YG(B)541D-II, Wenzhou Da Rong Textile Equipment Co. Ltd., Wenzhou, China). Breaking strengths of the cotton fabrics were measured according to ASTM D5035 (YG065HC, Lai Zhou Electronic Instrument Co., Ltd., Laizhou, China). Textile UV resistance property were measured according to AATCC-183-2014 (YG(B)912E, Wenzhou Da Rong Textile Equipment Co. Ltd., Wenzhou, China).

The antibacterial property of the cotton fabrics was tested according to GB/T 20944.3-2008 (DNP9272, ZHWY200, Shanghai Shidukai Instrument Equipment Co., Ltd., Shanghai, China). The bacteriostatic rate was calculated [Equation (5)] [20].
(5)$$Y(\%)=(1-\frac{B}{A})*100$$
where *Y* is the antibacterial rate, *A* is the number of colonies of the raw cotton fabric (CFU/mL), *B* is the number of colonies of the dyed cotton fabric (CFU/mL).

## 3. Results and Discussion

### 3.1. FTIR and K/S Values Analysis

The molecular structures of the raw, direct, mordant, and cross-linked dyed cotton fabrics were analyzed using FTIR spectra, the results are shown in Figure 2. The *K/S* values and digital images of the dyed cotton fabrics are shown in Table 1.

As shown in Figure 2, the absorption peak at 3410 cm^−1^ was assigned to –OH stretching vibration and the band at 2900 cm^−1^ was due to C–H stretching vibration [21]. It could be observed that all dyed cotton fabrics appeared a similar anthraquinone ring vibration peak of madder dye at 1597 cm^−1^ [2], indicating the madder dye could dye in the cotton fabrics. And all cross-linked dyed cotton fabrics appeared the ester carbonyl absorption peak near 1733 cm^−1^, showing that the esterification reaction occurred between dye-polycarboxylic acid-cotton fabric [22,23]. It could be seen from Table 1, the *K/S* value of the direct dyed cotton fabric was 2.24, indicating that the madder dye had a poor affinity for the cotton fabric. The *K/S* value of the alum mordant dyed cotton fabric was approximately 85% higher than that of the direct dyed cotton fabric, which indicated that the mordant dyeing enhanced the color strength of the madder dye on the cotton fabric. The *K/S* values of the four cross-linked dyed cotton fabrics were higher than that of mordant dyeing, indicating that the polycarboxylic acids as cross-linking agents further enhanced the color strength. The same conclusion could also be obtained from digital images of the dyed cotton fabrics. As shown in Table 1, compared with the direct dyed cotton fabric, the *K/S* values of the cross-linked dyed cotton fabrics increased more than 160%, revealing that the polycarboxylic acid not only reacted with the hydroxyl group of cellulose, but also had the esterification reaction with the hydroxyl group of dye. The polycarboxylic acid as cross-linking agent played a bridging role to connect with the dye and cotton fabric. The cross-linking process of dye-polycarboxylic acid-cotton fabric is shown in Figure 3.

Furthermore, the *K/S* value of the CA cross-linked dyed cotton fabric was lower than that of the other three compounds cross-linked dyed cotton fabrics. This is due to the fact that when CA was used as a single cross-linking agent, under the condition of catalytic heating, three carboxyl groups in the molecular structure of CA formed only two five-membered cyclic anhydrides intermediates successively and esterified two hydroxyl groups at most with low esterification sites. And in the same time, the α-hydroxyl group in the CA molecule structure hindered the progress of the cross-linking reaction, resulting in low esterification efficiency [14]. Therefore, the *K/S* value of the CA cross-linked dyed cotton fabric was relatively low. While when the mixtures of dicarboxylic acids and CA were used as cross-linking agents, firstly, the two adjacent carboxyl groups of one polycarboxylic acid generated five-membered cyclic anhydride intermediate under the condition of catalytic heating, and then reacted with the α-hydroxyl in the other polycarboxylic acid molecular structure to form a tetracarboxylic acid structure, at the same time, the close to the α-hydroxyl group and the negative effect on esterification reaction was reduced. Finally, the new tetracarboxylic acid formed three five-membered cyclic anhydrides intermediates successively and they could esterify three hydroxyl groups at most, so it increased the reaction sites and improved the efficiency of esterification cross-linking [15].

In addition, the SUA structure does not contain α-hydroxyl, the MLA structure contains one α-hydroxyl and the TTA structure contains two α-hydroxyl [24]. The experimental results showed that the α-hydroxyl hindered the esterification reaction, so the esterification reaction efficiency and the *K/S* value of the CA+SUA cross-linked dyed cotton fabric was the best, followed by CA+MLA, and CA+TTA was the lowest.

### 3.2. XRD Analysis

Crystal structures of the raw, direct, mordant, and cross-linked dyed cotton fabrics were analyzed by XRD and the crystallinities of the cotton fabrics were calculated by splitting and fitting the crystalline superimposed peaks and amorphous superimposed peaks of the diffraction curves with the Gaussian function [25]. The results are shown in Figure 4 and Table 2.

The dye solution can only spread into the amorphous and the edges of crystalline areas in the dyeing process. Figure 4 showed that the diffraction peaks of the raw and dyed cotton fabrics appeared all around 14.8°, 16.5°, 22.7°and 34.3°. These four diffraction peaks were formed, respectively, by (101), $\left(10\bar{1}\right)$, (002) and (040) of the cellulose crystal plane diffraction [26], and compared with the raw cotton fabric, the X-ray diffraction peak position of the dyed cotton fabrics had almost no change. It could be seen from Table 2 that compared with the raw cotton fabric, the crystallinities of the dyed cotton fabrics were changed approximately ±3%, indicating that dyeing processes had little effect for the crystallinity. XRD results showed that the direct, mordant and cross-linked dyeing mainly occurred on the active surface or amorphous areas, and retained the original crystal of the cotton fabrics [27].

### 3.3. Dyeing Property and Color Fastness Analysis

The color characteristic values, the hue angles (*h**°*), and color differences (Δ*E*) of the direct, mordant and cross-linked dyed fabrics were tested. These results are shown in Table 3.

As shown in Table 3, *L** corresponds to brightness (100 = white, 0 = black), the darker the surface color depth of the dyed cotton fabric, the less bright it is [28]. It could be seen from Table 1 that compared with the direct dyed cotton fabric, the *K/S* values of the mordant and cross-linked dyed cotton fabrics had been effectively improved, so the *L** values were reduced. The *a** value is the red-green coordinate (+ve = red, –ve = green), The *b** value is the yellow-blue coordinate (+ve = yellow, –ve = blue), and the *C** value is color saturation (100 = vivid, 0 = dull) [29,30]. Compared with the direct dyed cotton fabric, the *a**, *b** and *C** values of the mordant and cross-linked dyed cotton fabrics all increased, indicating that the color of the dyed cotton fabrics shifted to the red and yellow axis, and the color saturation increased. *h°* value is the chromaticity angle of the dyed cotton fabric, the range is 0~360° and 0/360–90° is defined as the red–yellow range, 90–180° is defined as the yellow–green range, 180–270° is defined as the green–blue range, 270–360/0° is defined as the blue–red range [31]. Table 3 showed the *h°* values of all dyed cotton fabrics were in the range of 0~90°, they were in the overlapping area of red and yellow, so the dyed cotton fabrics appeared orange.

The color levelness of the cotton fabric is evaluated according to the Δ*E* value. The Δ*E* value is less than 1, which means that the cotton fabric has great leveling property [32]. As shown in Table 3, the Δ*E* value of the direct dyed cotton fabric was 1.08, and it was more than 1, illustrating that its leveling property was poor. While the Δ*E* values of the mordant and cross-linked dyed cotton fabrics were less than 1, indicating their levelness properties were improved, and the Δ*E* value of the CA+SUA cross-linked dyed cotton fabric was the smallest (only 0.41), so its leveling property was the best. Since the madder dye has low molecular weight and poor affinity for the cotton fabric, the dye is only combined with cotton fabric through weak van der Waals in direct dyeing. Washing and soaping will easily cause the dye to fall off or shift on the surface, resulting in poor leveling property. The coordination bond formed by Al^3+^ ions can strengthen the bonding force between the dye and cotton fabric, and the leveling property was improved compared with the direct dyeing. During the cross-linked dyeing, the polycarboxylic acid and the dye were completely dissolved in the dye solution, and they were uniformly absorbed on the cotton fabric surface through the rolling method. At the same time, the strong chemical bonds between the dye and cotton fabric prevented the dye migrating and falling off during the curing and washing [33], therefore cross-linked dyeing further improved the leveling property of the cotton fabrics.

Washing, rubbing and light fastness were tested for the direct, mordant, and cross-linked dyed cotton fabrics, the results are given in Table 4.

It could be seen from Table 4 that compared with the direct dyed cotton fabric, the rubbing and washing fastness of the mordant and polycarboxylic acid cross-linked dyed cotton fabrics significantly improved. And the rubbing and washing fastness of the cross-linked dyed cotton fabrics increased by 4 levels or above. This can be attributed to the strong ester bonds of dye-polycarboxylic acid-cotton fabric, and in the process of rubbing and washing, the hydrolysis degree of the ester bonds was relatively little, so the rubbing and washing fastness obviously improved.

Natural dyes have poor light fastness. According to the reported results [5], under the visible light and ultraviolet light irradiation, the madder dye obviously fades on the cotton fabric. Therefore, it is particularly important to improve the light fastness of the madder dyed cotton fabric. As shown in Table 4, the light fastness of the direct dyed cotton fabric was only 1~2 levels, and the light fastness of the mordant dyed cotton fabric was not significantly improved. But the light fastness of the cross-linked dyed cotton fabric with polycarboxylic acid obviously increased, and in particular, the light fastness of CA+SUA cross-linked dyed cotton fabric reached 4~5 levels.

In order to intuitively study the light resistance of the dyed cotton fabrics, the dyed cotton fabrics were placed under a xenon arc lamp to simulate the intensity of sunlight at 12 noon, and the humidity and temperature of the room were maintained at 35% and 30 °C, respectively. The results of the *K/S* and *h°* values of the dyed cotton fabrics after 12 h and 24 h of light are shown in Table 5.

From Table 5, the direct and mordant dyed cotton fabrics began to fade after 12 h of exposure to simulated sunlight, and after 24 h, their *K/S* values decreased by 28.9% and 31.2%, *h°* values shifted to the yellow axis by 17.33° and 17.85°, respectively. And the color tonality of the direct and mordant cotton fabrics obviously changed. This is due to the fact that the madder natural dyes are easily oxidized and decomposed at exposure to sunlight, and the coordination bonds formed by Al^3+^ ions have poor light stability [5], leading to the poor light fastness of the direct and mordant dyed cotton fabrics. Furthermore, the structure of alizarin is decomposed to a variety of carboxyl compounds at sunlight condition, which increased the acidity of the dyed cotton fabrics. The alizarin structure appears yellow under acidic condition, so the colors of the dyed cotton fabrics were shifted to yellow after exposure to simulated sunlight. But the *K/S* and *h°* values of the cross-linked dyed cotton fabrics did not obviously change after 24 h of exposure to sunlight. They showed great light fastness, and the CA+SUA cross-linked dyed fabric had the best effect (4~5 levels). The great light resistance in the presence of the dyed cotton fabrics was explained by the fact that the diffusion of oxygen and moisture become difficult in the polycarboxylic acid cross-linked structure, which negatively affects the initiation of the photofading mechanism [13,34].

### 3.4. Wearing Property Analysis

Usually, the polycarboxylic acids are used as zero formaldehyde cross-linking agents to enhance the durable pressing property of the cotton fabric, so the wrinkle recovery angle and breaking strength of the raw, direct, mordant, and cross-linked dyed cotton fabrics were tested. The results are shown in Table 6.

Under stretching effect, macromolecular chains in the amorphous region of the cotton fabric are slipped, which causes the hydrogen bonds disassembly-reconstruction phenomenon. The newly formed hydrogen bonds prevent the macromolecular chains from returning to their original states and then forming irreversible wrinkles. It could be seen from Table 6 that compared with the raw cotton fabric, the sum of the warp and weft WRA of the direct and mordant dyed cotton fabrics did not obviously change, indicating that the direct and mordant dyeing did not improve the wrinkle resistance of the cotton fabrics. But the WRAs of the cross-linked dyed cotton fabrics obviously increased, this is due to the fact the esterification reaction occurred between dye-polycarboxylic acid-cotton fabric during the cross-linked dyeing process, leading to the formation of a cross-linked network structure on the surface of the cotton fabric prevented the slippage of the macromolecular chains [35,36]. Therefore, the wrinkle resistances of the cross-linked dyed cotton fabric were improved, and the CA+SUA cross-linked dyed cotton fabric increased by 40% than that of the raw cotton fabric, it had the best wrinkle resistance.

As shown in Table 6, compared with the raw cotton fabric, the breaking strengths of the direct and mordant dyed cotton fabrics had a smaller drop, both within 10%, while breaking strengths of the cross-linked dyed cotton fabrics obviously decreased, especially for the CA+SUA cross-linked dyed cotton fabric, the warp and weft breaking strengths were reduced by 22% and 21%, respectively. This is mainly due to the fact that the cellulose is not resistant to acid. Although the low concentration of polycarboxylic acid was used during the process of cross-linking dyeing, it also affected the breaking strength of the cotton fabric [37]. In addition, the curing of the cotton fabric at 160 °C resulted in a certain loss of strength [36].

UV radiation refers to the irradiation of two wavelength bands, including UVA (320–400 nm) and UVB (280–320 nm). Long-term exposure to UV radiation will cause skin aging and carcinogenesis [38]. In order to evaluate the UV protective property of the raw and dyed cotton fabrics, the UV transmittances in the ultraviolet region (280~400 nm) and UPF values were tested, and the results are shown in Figure 5 and Table 7.

Figure 5 and Table 7 showed that more than 50% of ultraviolet light passed through the raw cotton fabric, and its UPF value was only 13.35, but compared with the raw cotton fabric, the UV transmittances of the dyed cotton fabrics were greatly reduced and their UPF values were obviously improved. This reason is that the aromatic ring and conjugated structure of natural madder dye has a strong UV absorption ability. It also could be seen from Figure 5 and Table 7 that the average transmittance of the compound cross-linked dyed cotton fabrics in the UVA region were about 5%, and the UPF values were greater than 50. According to the requirements of AATCC-183-2014, when the UPF value > 40 and UVA < 5%, the fabrics have excellent UV resistance. The reason is that more dyes in the cotton fabrics had a better ability to absorb ultraviolet light [5]. At the same time, the cross-linked network structure could effectively prevent the passage of ultraviolet light and reduce the transmittance of UV. The combination of the two effects made the cross-linked dyed cotton fabric had great UV resistance property.

In general, the natural dyes have certain antibacterial property [39], so the antibacterial activities of the dyed cotton fabrics with madder had been tested. The *E. coli* (Gram-negative) and *S. aureus* (Gram-positive) as representative microorganisms were chosen in this study, and the results are shown in Figure 6.

Figure 6 showed that the raw cotton fabric had no antibacterial activity for both *S. aureus* and *E. coli*. The antibacterial property of the dyed cotton fabrics was significantly improved compared with the raw cotton fabric. In particular, the antibacterial activity of the compound cross-linked dyed cotton fabrics increased by more than 60%. This is mainly due to the fact the anthraquinone structure of the madder dye was found to complex irreversibly with nucleophilic amino acids in proteins, leading to inactivation of the protein and loss of function, and hence it could inhibit the growth of both Gram-positive and Gram-negative bacteria [39]. The SUA+CA cross-linked dyed cotton fabric had the highest color yield, so it had the best antibacterial property (about 70%).

## 4. Conclusions

In this study, CA along with compounds of CA+TTA, CA+MLA, CA+SUA were used as cross-linking agents for dyeing of the cotton fabric with madder dye. FTIR showed that the ester bonds cross-linking was formed between dye-polycarboxylic acid-cotton fabric, and XRD indicated the crystal structures of the cotton fabrics were not affected after cross-linkage. Compared with traditional direct and mordant dyed cotton fabrics, the *K/S* values, levelness properties, washing, rubbing and light fastness, WRAs, and UV protective properties of the cross-linked dyed cotton fabrics were significantly improved. Among them, CA+SUA cross-linked dyed cotton fabric obtained the best dyeing effect. Its *K/S* value was 8.37, and the washing, rubbing and light fastness were all 4~5 levels. Also, the CA+SUA cross-linked dyed cotton fabric achieved great anti-wrinkle and UV protective property and exhibited good antibacterial property. Therefore, the polycarboxylic acids as cross-linking agents were more suitable for the dyeing on cotton fabric with the madder dye. It can not only improve the dyeing property of the cotton fabric, but also give the cotton fabric excellent wrinkle resistance and UV resistance.

## Figures and Tables

**Figure 1 polymers-13-03289-f001:**
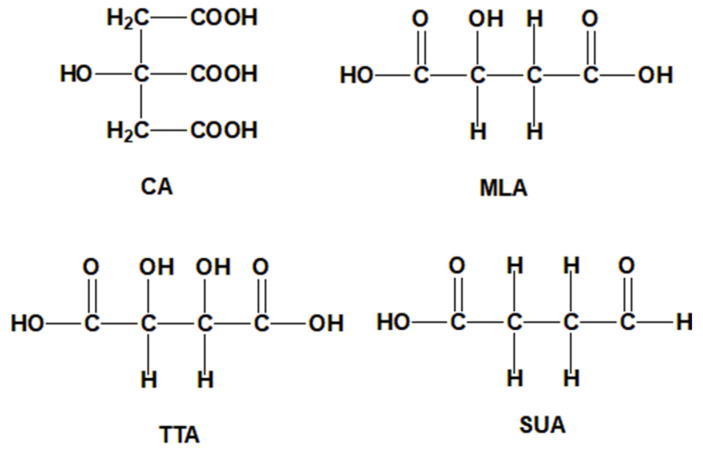
Structures of the polycarboxylic acids.

**Figure 2 polymers-13-03289-f002:**
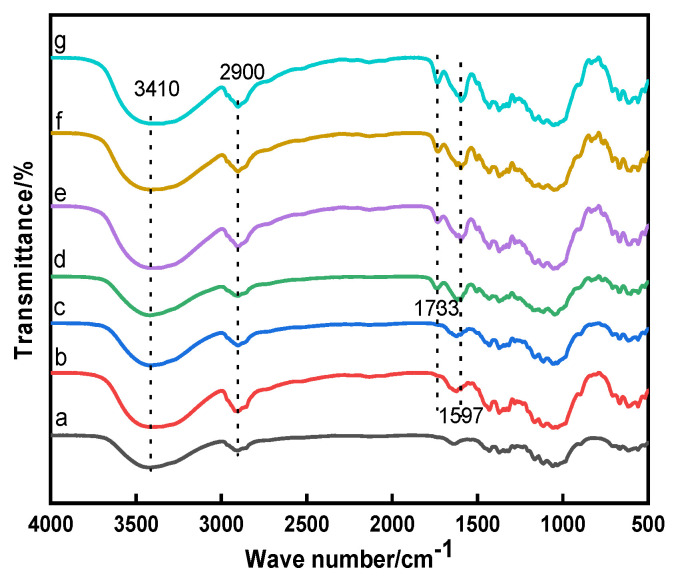
FTIR spectra of the cotton fabrics: a, Raw; b, Direct; c, Mordant; d, CA; e, CA+TTA; f, CA+MLA; g, CA+SUA.

**Figure 3 polymers-13-03289-f003:**
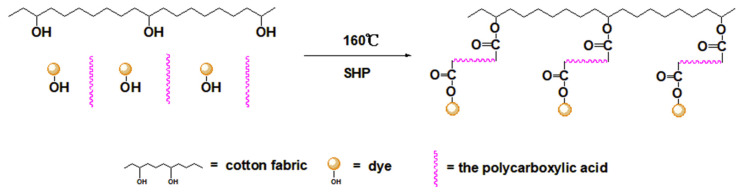
Schematic diagram of the polycarboxylic acid cross-linking process.

**Figure 4 polymers-13-03289-f004:**
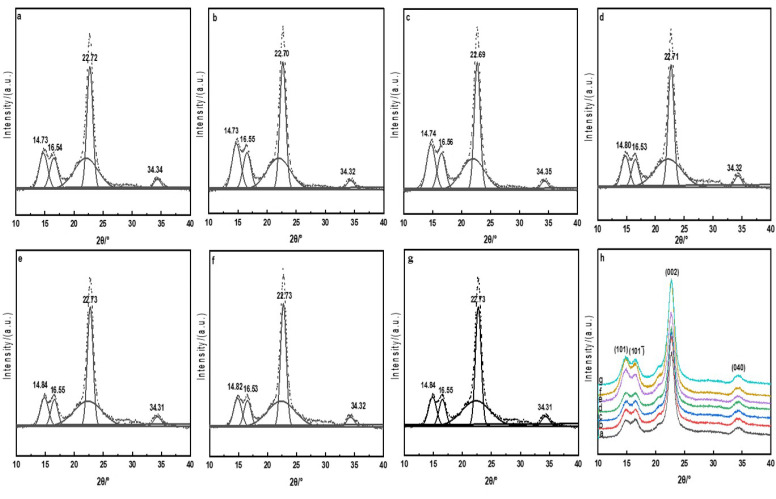
XRD peak-splitting fitting diagrams of the cotton fabrics (**a**, raw; **b**, Direct; **c**, Mordant; **d**, CA; **e**, CA+TTA; **f**, CA+MLA; **g**, CA+SUA c), **h**, X-ray diffraction curves of the cotton fabrics.

**Figure 5 polymers-13-03289-f005:**
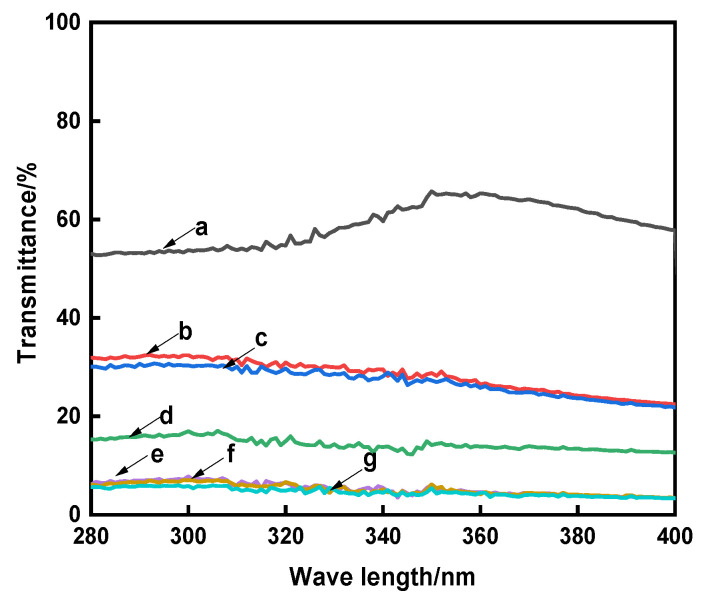
The UV transmittances of the cotton fabrics (a, Raw; b, Direct; c, Mordant; d, CA; e, CA+TTA; f, CA+MLA; g, CA+SUA).

**Figure 6 polymers-13-03289-f006:**
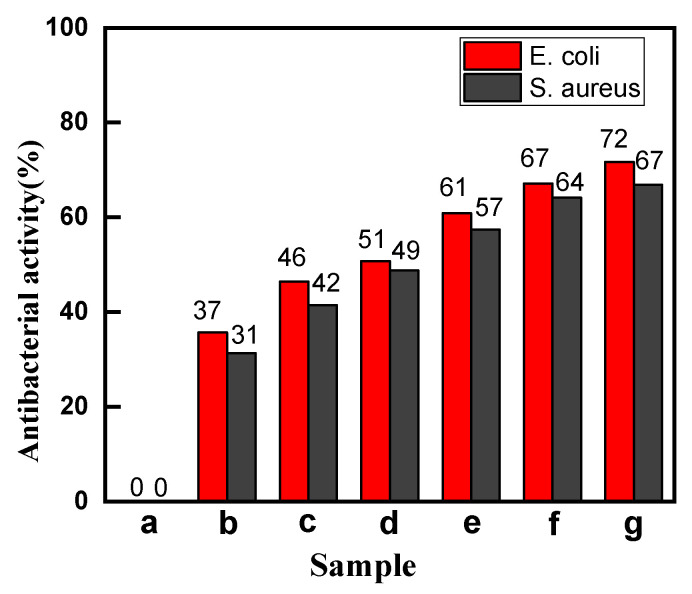
The antibacterial rates of the cotton fabrics (a, Raw; b, Direct; c, Mordant; d, CA; e, CA+TTA; f, CA+MLA; g, CA+SUA).

**Table 1 polymers-13-03289-t001:** The *K/S* values and digital images of the dyed cotton fabrics.

Sample	The *K/S* Value	Digital Images
Direct	2.24	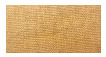
Mordant	4.15~85%	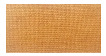
CA	5.88~160%	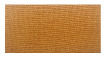
CA+TTA	6.55~190%	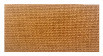
CA+MLA	7.53~240%	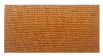
CA+SUA	8.37~270%	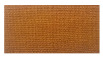

Note: The color of the raw cotton fabric is pure white.

**Table 2 polymers-13-03289-t002:** Fitting data and crystallinity of the cotton fabrics.

Sample	Crystal Peak Area				Amorphous Peak Area	Crystallinity/%
	1	2	3	4		
Raw	2525	2133	6126	610	6118	65.06
Direct	3579	2633	6817	608	6528	67.63
Mordant	3329	2124	5316	486	5780	66.07
CA	1706	1385	4834	555	5163	62.16
CA+TTA	1703	1364	5046	549	5239	62.31
CA+MLA	1678	1386	5221	557	5300	62.52
CA+SUA	1493	1257	4828	539	5018	61.80

**Table 3 polymers-13-03289-t003:** Color characteristics, *h°* values and Δ*E* values of the direct, mordant and cross-linked dyed cotton fabrics.

Sample	*L**	*a**	*b**	*C**	*h°*	Δ*E*
Direct	72.41	28.61	42.91	49.76	56.29	1.08
Mordant	63.49	24.02	43.54	51.55	61.07	0.73
CA	61.41	24.66	45.27	51.81	61.41	0.57
CA+ TTA	57.53	25.33	44.79	51.97	60.51	0.55
CA+ MLA	56.85	27.44	44.99	52.84	58.61	0.53
CA+ SUA	55.51	27.74	45.11	53.09	58.41	0.41

Note: “*” was representation methods of the CIE color measurement system in 1976.

**Table 4 polymers-13-03289-t004:** Color fastness of the dyed cotton fabrics.

Sample	Rubbing Fastness		Washing Fastness	Washing Fastness to Staining					Light Fastness
	Dry	Wet		Cotton	Wool	Acrylic	Dacron	Chinlon	
Direct	2~3	2	2	3	2~3	2~3	3	3	1~2
Mordant	3~4	3~4	3	4	4	4	4	4	2~3
CA	4~5	4	4	4~5	4~5	4	4~5	4~5	3~4
CA+TTA	4~5	4	4	4~5	4~5	4	4~5	4~5	4
CA+MLA	4~5	4	4	4~5	4~5	4	4~5	4~5	4
CA+SUA	4~5	4~5	4~5	4~5	4~5	4~5	4~5	4~5	4~5

**Table 5 polymers-13-03289-t005:** The *K/S* and *h°* values of the dyed cotton fabrics at sunlight exposure 12 h and 24 h.

Sample	Sunlight Exposure 12 h				Sunlight Exposure 24 h			
	*K/S* Value	Δ*K/S* Value	*h°* Value	Δ*H°* Value	*K/S* Value	Δ*K/S* Value	*h°* Value	Δ*h°* Value
Direct	1.89	−0.35	60.09	+3.8	1.36	−0.88	73.62	+17.33
Mordant	3.89	−0.26	64.92	+3.85	3.61	−0.54	78.92	+17.85
CA	5.79	−0.09	62.10	+0.69	5.64	−0.24	66.10	+4.69
CA+TTA	6.48	−0.07	61.21	+0.7	6.31	−0.24	65.24	+4.73
CA+MLA	7.46	−0.07	59.30	+0.69	7.39	−0.14	62.80	+4.19
CA+SUA	8.31	−0.06	59.04	+0.63	8.23	−0.14	62.53	+4.12

Note: The initial *K/S* and *h°* values of the dyed cotton fabrics are shown in Table 1 and Table 3, respectively.

**Table 6 polymers-13-03289-t006:** Wearing property of the raw, direct, mordant, and cross-linked dyed cotton fabrics.

Sample	WRA/° (W+F)	Breaking Strength/N	
		Warp	Weft
Raw	137.8	329	324
Direct	138.1	316—4%	313—3%
Mordant	139.3—1%	309—6%	311—4%
CA	164.6—19%	288—12%	285—12%
CA+TTA	172.9—25%	277—16%	278—14%
CA+MLA	184.5—34%	269—18%	264—19%
CA+SUA	193.4—40%	257—22%	257—21%

**Table 7 polymers-13-03289-t007:** The UPF values of the raw, direct, mordant, and cross-linked dyed cotton fabrics.

Sample	Raw	Direct	Mordant	CA	CA+TTA	CA+MLA	CA+SUA
UPF value	13.35	27.54	36.31	47.39	50+	50+	50+

## Data Availability

The data presented in this study are available on request from the corresponding author.

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
