# Peer review of "Dyeing Property Improvement of Madder with Polycarboxylic Acid for Cotton"

_polymers, 2021, doi:10.3390/polym13193289_

Round 1

Reviewer 1 Report

The manuscript “Dyeing property improvement of madder with polycarboxylic acid for cotton” summarized the potential applications of polycarboxylic acid for cotton dyeing with natural dye. This manuscript is generally well written. I think it will attract the attention of the readers. It can be published on this journal after dressing the following comments

  1. plagiarism is 4 % see attached file 
  1. Normally citric acid provides good cross-linking properties, so why do you use it in combination with other dicarboxylic acids, polymerization can occur with citric acid as it contains hydroxyl and carboxyl groups and provides more carboxyl groups after polymerization
  2. They provide that the dyed fabric can be protected from UV by addressing the UPF values, I suggest this measurement to improve his explanation

Author Response

1 plagiarism is 4 % see attached file:

Response1: I had already understood the attachment carefully, thank you.

2 Normally citric acid provides good cross-linking properties, so why do you use it in combination with other dicarboxylic acids, polymerization can occur with citric acid as it contains hydroxyl and carboxyl groups and provides more carboxyl groups after polymerization.

Response2: Because the molecular structure of CA contained three carboxyl groups, its esterification efficiency is lower than that of the tetracarboxylic acid, but BTCA is expensive. And the CA and dicarboxylic acids formed a tetracarboxylic acid structure under the catalytic heating condition, its improved esterification efficiency so I used CA in combination with other dicarboxylic acids.

And then, the self-polymerization of citric acid required polyols as monomers and harsh experimental conditions, self-polymerization and dyeing needed to be completed, separately. The entire experiment required multiple steps and it was cumbersome.

3 They provide that the dyed fabric can be protected from UV by addressing the UPF values, I suggest this measurement to improve his explanation

Response3: Thank you for your commendations. In this manuscript, the UPF values of the cotton fabrics further were measured on the basis of measuring the UV transmittances. It could be seen in Table 7(in 3.4, lines of 345-367) in the manuscript for details. Thank you.

Reviewer 2 Report

This manuscript is written well. I think it is attract the attention to the readers. It can be published on this journal after reviesed the following comments

  1. why authors use citric acid in combination with dicarboxylic acids (even some of them didn't provides hydroxyl group to contain the polymerisation)
  1. One of the important properties for dyed fabric with natural dye is the antimicrobial properties and the author didn’t provide anything about it, so the author should provide the antimicrobial properties for treated fabric with comparing with untreated one

Author Response

1 why authors use citric acid in combination with dicarboxylic acids (even some of them didn't provides hydroxyl group to contain the polymerisation)

Response1: The reason why I use the combination of citric acid and dicarboxylic acid has been explained in the introduction (lines of 69-75) and FTIR and K/S values analysis (in 3.1, lines of 194-208) part in the manuscript.

The purpose of choosing these three dicarboxylic acids was to explore whether the hydroxyl groups of the dicarboxylic acids had an effect on the complex cross-linking. The experimental results showed that this effect existed. That is, the hydroxyl group on TTA and MLA might have a negative effect on the esterification reaction, so SUA+CA compound cross-linking effect was best.

2 One of the important properties for dyed fabric with natural dye is the antimicrobial properties and the author didn’t provide anything about it, so the author should provide the antimicrobial properties for treated fabric with comparing with untreated one

Response2: Thank you for your commendations. Antibacterial properties of the raw and dyed cotton fabrics were measured in my manuscript. The antibacterial property analysis and results showed in Figure 6(in 3.4, lines of 369-384) in the manuscript.

Reviewer 3 Report

The work "Dyeing property improvement of madder with polycarboxylic acid for cotton" by Hong Li et al the authors analyze some new recipes to improve the mordant of natural madder dyes on cotton.

For these they use a mixture of polycarboxylic acids. This, however, is the only polymeric starting point of the entire work.

The work is interesting and worthy of publication, perhaps, but I think "Polymers" is not the most suitable Jounal. "Molecules" would probably be more appropriate (Special Issue "Exclusive Feature Papers in Colorants")

Abbreviations should be made clear the first time they appear (e.g. K/s in the abstract, etc.).

Table 1 is useless, the data contained are extensively discussed in the text a few lines earlier.

It would be more appropriate to delete figure 4. The textual description is more than sufficient. Furthermore, no analysis (or experimental investigation) "in this article" seems to me to have been performed to corroborate this hypothesis.

Author Response

1 The work "Dyeing property improvement of madder with polycarboxylic acid for cotton" by Hong Li et al the authors analyze some new recipes to improve the mordant of natural madder dyes on cotton. For these they use a mixture of polycarboxylic acids. This, however, is the only polymeric starting point of the entire work. The work is interesting and worthy of publication, perhaps, but I think "Polymers" is not the most suitable Jounal. "Molecules" would probably be more appropriate (Special Issue "Exclusive Feature Papers in Colorants")

Abbreviations should be made clear the first time they appear (e.g. K/s in the abstract, etc.).

Response1: Thank you for your suggestion, I had revised it in the manuscript.

2 Table 1 is useless, the data contained are extensively discussed in the text a few lines earlier.

Response2: Thank you for your suggestion, Table 1 had been deleted from my manuscript, thank you.

3 It would be more appropriate to delete figure 4. The textual description is more than sufficient. Furthermore, no analysis (or experimental investigation) "in this article" seems to me to have been performed to corroborate this hypothesis.

Response3: I had deleted Figure 4 in the manuscript. Thank you.

Reviewer 4 Report

It is a very good study with overall adequate presentation of experimental results. Some additions are needed:

1) Authors should further emphasize on the novelty of their work.

2) Some minor typos, grammar and syntax errors should be carefully revised and corrected accordingly.

3) Reference can be even more updated (more recent relative works).

Author Response

1 Authors should further emphasize on the novelty of their work.

Response1: In this study, the polycarboxylic acids as cross-linking agents were more suitable for the dyeing on cotton fabric with the madder dye. It could not only improve the dyeing property of the cotton fabric, but also get multifunctional cotton fabric textiles. This was the novelty of their work, and it had been reflected in the introduction(lines of 68-69, lines of 79-81) and conclusion(lines of 396-399) of the article.

2 Some minor typos, grammar and syntax errors should be carefully revised and corrected accordingly.

Response2: I had re-checked the grammar with other English teachers. Thank you.

3 Reference can be even more updated (more recent relative works).

Response3: Thank you for your comments, I had revised some of the references (references 14,15,16 were replaced), thank you.

Round 2

Reviewer 4 Report

All my comments of the initial submission have been correctly replied and included in the revised manuscript. The quality of this work has been drastically improved after revision and therefore I recommend its publication as it is.